# Multilocus Phylogeography and Population Genetic Analyses of *Opsariichthys hainanensis* Reveal Pleistocene Isolation Followed by High Gene Flow around the Gulf of Tonkin

**DOI:** 10.3390/genes13101908

**Published:** 2022-10-20

**Authors:** Junjie Wang, Wenjun Zhang, Jinxian Wu, Chao Li, Yu-Min Ju, Hung-Du Lin, Jun Zhao

**Affiliations:** 1Guangdong Provincial Key Laboratory for Healthy and Safe Aquaculture, Guangdong Provincial Engineering Technology Research Center for Environmentally-Friendly Aquaculture, Guangzhou Key Laboratory of Subtropical Biodiversity and Biomonitoring, School of Life Science, South China Normal University, Guangzhou 510631, China; 2National Museum of Marine Biology and Aquarium, Pingtung 94401, Taiwan; 3Department of Marine Biotechnology and Resources, National Sun Yat-sen University, Kaohsiung 80424, Taiwan; 4The Affiliated School of National Tainan First Senior High School, Tainan 701, Taiwan

**Keywords:** DIY-ABC, Hainan Island, *Opsariichthys hainanesis*, phylogeography

## Abstract

The ichthyofauna of continental islands is characterized by immigration through a land bridge due to fluctuating sea levels. Hainan Island is adjacent to the southern margin of mainland China and provides opportunities for understanding the origin and diversification of freshwater fishes. The aim of our study was to evaluate the level of genetic variation and phylogeographic structure of *Opsariichthys hainanensis* on Hainan Island and mainland China, using mtDNA cyt *b* gene (1140 bp) and D-loop (926 bp), nuclear RAG1 gene (1506 bp), and 12 microsatellite loci. Mitochondrial phylogenetic analysis identified five major lineages according to the geographical distribution from different populations. We suggested that two dispersal events occurred: the population in the Changhua River migrated to the Red River (Lineage B), and the populations in the South Hainan region moved northwards to the North Hainan region. However, populations in Northwest Hainan Island dispersed to the populations around the Gulf of Tonkin (Lineage A1) and populations in Northeast Hainan Island dispersed to the populations in mainland China (Lineage A2). Our results indicated that the populations of *O. hainanensis* suffered a bottleneck event followed by a recent population expansion supported by the ABC analysis. We suggest that *O. hainanensis* populations were found mostly in the lowlands and a lack of suitable freshwater habitat in southern mainland China and Hainan during the Last Interglacial period, and then expansion occurred during the Last Glacial Maximum.

## 1. Introduction

Continental islands, due to their close proximity to neighboring continental areas, provide excellent opportunities for understanding the origin and diversification of freshwater fishes [1]. During glacial phases, the islands and the mainland were connected by land bridges due to the decrease in sea level. Many studies provide evidence supporting continental islands receiving their freshwater fishes directly from the adjacent mainland [2]. Adjacent to the southern margin of mainland China are numerous continental shelf islands, and the islands of Taiwan and Hainan present the two largest islands. Hainan Island is slightly smaller in size than Taiwan but contains a higher diversity of freshwater fishes than Taiwan. The high degree of diversity for the freshwater fishes of Hainan compared with Taiwan Island reflects the weak effect of geographical isolation due to the Qiongzhou Strait, which is approximately 70 km long and 30 km wide, with a maximum depth of 120 m, and Hainan is separated from mainland China. Previous studies on the phylogeography of freshwater fishes between the island and mainland have focused on Taiwan and mainland China (e.g., *Onychostoma barbatulus* [3]; *Squalidus argentatus* [2]; *Cobitis sinensis* [4]; *Opsariichthys* [5]; *Rhodeus ocellatus* [6]; *Acrossocheilus paradoxus* [7]). There is relatively little information available on the phylogeography of freshwater fishes between Hainan Island and mainland China (e.g., *Garra orientalis* [8]; *Aphyocypris normalis* [9]; *O. hainanensis* [10]; *Channa gachua* [1]).

Geological events have strongly influenced the distribution and migration of primary freshwater fishes. Therefore, the distribution of primary freshwater fishes often has obvious zoogeographical fauna [11]. Hainan Island does provide opportunities for geographic isolation of migration routes. Hainan Island has a staircase-like topographic structure, with the highest mountain, Wuzhishan Mountain and Yinggeling Mountain Range (WY Range) (1876 m), standing at its center. The topology of the island descends step by step from towering mountains to flat tablelands and plains, rises steeply from the southern regions and extends north to a wide plain. Rivers originate mostly from the central mountainous area, forming a radiating river system (e.g., the four largest rivers are the Nandu, Changhua, Wanquan, and Linshui Rivers). Hainan Island has experienced complicated historical events (Figure 1). The uplift of the continental shelf during the ice age also created many opportunities for water systems to converge in coastal estuaries, and the confluence of water systems also provided opportunities for biological diffusion [5,8] (Lin et al., 2016; Yang et al., 2016). In previous phylogeographic studies of freshwater fishes on Hainan Island, freshwater fishes migrated from mainland China and Vietnam via the Qiongzhou Strait and the Gulf of Tonkin during Pleistocene glaciations [1,8]. During the ice age, the Gulf of Tonkin was exposed because of the drop in sea level, and the entire area, including the Gulf of Tonkin and Hainan Island, became part of the coastal plains of the Asian continent. The northern water system of Vietnam flows into the South China Sea through the Leizhou Peninsula and the southwestern part of Hainan Island [1,12,13]. There are 154 rivers flowing into the sea on Hainan Island, and previous phylogeographic studies have focused on the major river system. There is little information about the role played by the smaller river systems in northwestern Hainan.

On Hainan Island, a hotspot of freshwater fish diversity has been severely reduced due to overexploitation, water pollution, river flow modification, and destruction of habitat [14]. Identifying patterns and drivers of genetic diversity in freshwater fishes provides a valuable additional tool for conservation managers to predict how species might adapt and respond to continued global change [15]. *O. hainanesis* (Nichols and Pope 1927) belongs to the order Cypriniformes, family Cyprinidae, and genus *Opsariichthys* and is a small cyprinid fish found in running streams, while adult individuals inhabit medium-sized to large rivers distributed on Hainan Island and the southern region of Yunkai Mountain in mainland China [5]. In a previous study, Zhang et al. [10] studied the phylogeography of the same fish using the mitochondrial cytochrome *b* (cyt *b*) gene, but insufficient sampling density and genetic markers were available to clarify the phylogeography of this species on Hainan Island and mainland China. In the present study, we used the mitochondrial DNA (mtDNA) cytochrome *b* gene (cyt *b*) and control region (D-loop), nuclear gene rag 1 region (RAG 1), and 12 microsatellite DNA markers to establish the phylogeography and genetic structure in Hainan and mainland China. There are three major questions in our study: (1) What is the level of genetic diversity and distribution of genetic variation in *O. hainanensis*? (2) How and when did *O. hainanensis* colonize the rivers of different geographical districts on Hainan Island and mainland China? (3) Was the broadly distributed *O. hainanensis* affected by the appearance of geographical barriers during the Last Glacial Maximum (LGM). The results may provide functional insight into the phylogeography and population structure of *O. hainanensis* that make them useful for sustainable river management and conservation of freshwater fishes on Hainan Island and mainland China.

## 2. Materials and Methods

### 2.1. Sample Collection, Microsatellite Genotyping, and Mitochondrial Sequencing

A total of 305 samples of *O. hainanensis* were collected from twenty-one populations on Hainan Island and mainland China during 2017 and 2019 (Table 1, Figure 1). The populations belonged to three regions: one population in Red River (Yuanjiang River, HH), fourteen populations on Hainan Island (Changhua River, CH; Nandu River, ND; Wanqan River, WQ; Lingshui River, LS; Tengqiao River; TQ, Baisha River, BS; Wanglou River, WL; Chunjiang River, CJ; Beimen River, BM; Zhubi River, ZB; Wenchang River, WC; Longgun River, LG; Longshou River, LOS; Longwei River, LW), and six populations in mainland China (Tan River, TR; Jian River, JR; Moyang River, MY; Nanliu River, NL; Fangheng River, FC; Beilun River, BL) (Table 1 and Figure 1). Locality information and sample numbers are provided in Table 1. All animal studies were conducted in accordance with the guidelines and approval of the Animal Research and Ethics Committee of School of Life Science, South China Normal University (permissions, CAMC-2018F). A piece of muscle tissue below the dorsal fin was obtained and preserved in 95% alcohol and frozen at −20 °C for DNA extraction after the morphological identification of all individuals.

Total genomic DNA was extracted for each sample using the DNAeasy Blood and Tissue Kit (QIAGEN, Shenzhen, China) following the manufacturer’s protocol. The mitochondrial complete cyt *b* gene and control region (D-loop) were amplified by polymerase chain reaction (PCR) using the universal primers L14724 and H15915 [16] and tRNA-PHE (5′-AAAGCATCGGTCTTGTAATCCGAAG-3′) and 12S rRNA (5′-CATGCGGAGTTTCTTAGGTC-3′) [1], respectively. The nuclear gene (RAG1) was amplified using primers RAG1F1 (5′-CTGAGCTGCAGTCAGTACCATAAGATGT-3′) and RAG1R1 (5′-CTGAGTCCTTGTGAGCTTCCATRAAYTT-3′) [17]. PCR amplification and sequencing were performed as described by Zhang et al. [10]. Microsatellite sequences were isolated from the genomic DNA of one *O. hainanensis* individual from the Changhua River on Hainan Island using the Illumina HiSeq X Ten platform (Illumina, Shanghai, China) to provide a database of resulting microsatellites. After Raw Illumina reads filtering adapter, low-quality and duplicate reads, the clean reads were comparatively analyzed using SOAP denovo software (http://soap.genomics.org.cn/soapdenovo.html, accessed on 1 June 2022). Simple repeat sequences (SSRs) were screened from the merged sequences using MISA (http://pgrc.ipk-gater slebe n.de/misa/, accessed on 1 June 2022), accepting dinucleotide repeats of ≥8, trinucleotide repeats of ≥6, and tetranucleotide repeats of ≥4. The PRIMER3 program was used for each microsatellite locus in the flanking region with a product of 100–500 bp to design PCR primers for amplification for each region. A total of 40 randomly selected primer pairs were validated in five individuals, with 12 of the primer pairs amplifying a consistent product. To confirm the repeat motifs per locus, all amplified products of the 12 loci were sequenced using an ABI 3730xl DNA Analyzer (Applied Biosystems, Waltham, MA, USA). The selected primers were synthesized with M13 tailed forward primers (5′-TGTAAAACGACGGCCAGT-3′) labeled with one of the following fluorescent dyes: fluorescein amidites (FAM), hexachlorofluorescein (HEX), or carboxyrhodamine (ROX). A total of 30 samples of *O. hainanensis* were collected from the Nandu River (ND) on Hainan Island and the Moyangjiang River (MY) in mainland China to test the polymorphism of loci. PCR was performed under the following conditions: 95 °C for 5 min, 35 cycles at 95 °C for 30 s, at the annealing temperature of each primer (Appendix A) for 30 s, an extension at 72 °C for 30 s, and a final 10 min extension at 72 °C. The fluorescent products were detected by capillary electrophoresis using an ABI 3730xl DNA Analyzer (Applied Biosystems), and the data were obtained with GeneMapper v4.0 software (Applied Biosystems, USA). All nucleotide sequences were deposited in GenBank under accession numbers MN325030-MN325066.

### 2.2. Data Analysis

#### 2.2.1. Mitochondrial DNA and Nuclear DNA Analysis

Two mtDNA genes (cyt *b* gene and d-loop region) and one nuDNA gene (RAG1) were aligned using Clustal X 2.0 software [18]. The indices of the number of haplotypes (Nh), haplotype diversity (h), nucleotide diversity (current genetic diversity estimates (θπ), and historical diversity estimates (θω); [19]) were calculated using DnaSP v5.0 software [20]. Comparing the current genetic diversity estimates (θπ) and historical diversity estimates (θω) provided insights into the evolution and population dynamics over recent evolutionary history [21]. We reconstructed phylogenetic trees by three methods, including MEGA X for neighbor-joining (NJ) [22], the PhyML v.3.0 web server with maximum likelihood (ML) [23], and MrBayes v. 3.2.6 for Bayesian inference (BI) [24]. We used the Akaike Information Criterion (AIC) in PhyML with Smart Model Selection (http://www.atgc-montpellier.fr/phyml-sms/, accessed on 1 June 2022) [25] for selection of the best-fit nucleotide substitution model. A minimum spanning network of haplotypes was drawn using the minimum spanning network method (minspnet in Arlequin 3.5) [26]. In addition, we estimate a time period for the most recent common ancestor (TMRCA) for each lineage as implemented in BEAST v1.8.2 software [27]. In this study, Divergence times were estimated under a strict molecular clock (uncorrelated lognormal), and mutation rates for D-loop and cyt *b* were regarded as 3.6% and 0.76% per million years in cyprinid fishes, respectively [28,29].

To determine the scenarios of demographic expansion, we performed neutrality tests (Tajima’s D test [30] and Fu’s *F*s test [31]) and mismatch distributions using DnaSP v5.0 software [20]. Furthermore, Bayesian skyline plots were generated for two mtDNA genes to determine the effective population size changes over time using BEAST v1.8.2 [27]. Skyline plots run for >200,000,000 iterations each to ensure convergence of all parameters (ESSs > 200), with the first 10% of samples for each chain discarded as burnin and then drawn using Tracer v1.6 [32]. In this study, mutation rates for D-loop and cyt *b* were regarded as 3.6% and 0.76% per million years in cyprinid fishes for population expansion, respectively [28,29].

Population genetic structures were performed with pairwise *F*_ST_ values and a hierarchical analysis of molecular variance (AMOVA) using Arlequin 3.5 [26], followed by statistical significance with 10,000 permutation steps for each comparison. We then used four scenarios according to geographical barriers to construct a hierarchical cluster analysis in AMOVA: (1) Scenario Ⅰ: two independent groups including the island group (CH, ND, WQ, LS, TQ, BS, WL, CJ, BM, ZB, WC, LG, LOS, LW) and the mainland group (HH, TR, JR, MY, NL, FC, BL), which were primarily divided by the Qiongzhou Strait; (2) Scenario Ⅱ: three independent groups including the Red river group (HH), the island group (CH, ND, WQ, LS, TQ, BS, WL, CJ, BM, ZB, WC, LG, LOS, LW), and the mainland group (TR, JR, MY, NL, FC, BL), which were primarily divided by the Qiongzhou Strait and Beibu Gulf; and (3) Scenario Ⅲ: four groups including the Red river group (HH), the North Hainan island group (ND, WQ, CJ, BM, ZB, WC, LG, LOS, LW), the South Hainan island group (CH, LS, TQ, BS, WL), and the mainland group (TR, JR, MY, NL, FC, BL), which were primarily divided by the WY Range, the Qiongzhou Strait, and Beibu Gulf. The program SAMOVA was used to explore the population structure of sampling areas with the maximum extent of genetic differentiation of *O. hainanensis* [33]. We performed these analyses based on 1000 simulated annealing steps and compared maximum indicators of differentiation (*F*_CT_) when the program was instructed to identify K = 2 through K = 15 partitions of the sampling area. Ancestral areas were reconstructed using Bayesian binary MCMC (BBM) with default parameters for *O. hainanensis* using RASP 3.2 [34]. The sampling and distribution populations of *O. hainanensis* were defined for the biogeographic analyses.

#### 2.2.2. Microsatellite DNA Analysis

The software program MICROCHECKER was employed to infer the most likely technical cause of Hardy Weinberg Equilibrium (HWE) departures, including null alleles, allelic dropouts due to short allele dominance, and errors made during the scoring of alleles with ‘stutter’ in our data [35]. Based on microsatellite data, the number of alleles (Na), mean observed heterozygosity (H_O_), mean expected heterozygosity (H_E_), deviations from Hardy Weinberg expectations (HWEs), and tests of linkage disequilibrium between all pairs of loci within locations and overall were calculated with Arlequin v3.5 [26]. Allelic richness (A_R_) and inbreeding coefficient (*F*_IS_) [36] were estimated using FSTAT software for Windows v2.9.3 [37]. The population genetic differentiation of *O. hainanensis* (pairwise *F*_ST_ and *R*_ST_ values) was performed with the same software and procedures as those used for mitochondrial DNA data (AMOVA and pairwise *F*_ST_ mentioned above).

Bayesian assignment tests were applied to estimate the number of genetic clusters and to evaluate the degree of admixture among these clusters using STRUCTURE v2.3.3 [38] based on microsatellites. An estimation of the number of subpopulations (K) was completed using 20 independent runs with K = 2–15 (assuming no prior population delineation information) at 100,000 MCMC repetitions combined with a 10,000 repetition burn-in period. The posterior probability was calculated for each value of K using the estimated log-likelihood of K, and a likelihood ratio test was used to determine the optimal number of subpopulations [38]. The most likely K-value was determined in Structure Harvester Web 0.6.94 [39]. Furthermore, a principal component analysis (PCA) was performed via GenAlEx version 6.503 [40] based on the standardized covariance of genetic distances between populations, which revealed the genetic relationships for the geographic region of allelic divergence between populations of *O. hainanensis* in multivariate space. To explore the relationships among the populations, the Cavalli-Sforza and Edwards genetic distance (Dc) between all pairs of populations was calculated, and a dendrogram was also created by the neighbor-joining methods calculated in POPULATIONS ver. 1.2.28 [41]. To detect whether the *O. hainanensis* populations have experienced a recent reduction in the effective population size, the software program Bottleneck 1.2.02 [42] was used. The observed allele frequency distribution was compared with the frequency distribution of a population in mutation-drift equilibrium assuming the infinite allele model (IAM), stepwise mutation model (SMM), and two-phase model (TPM), which consists of a 70% stepwise mutation model and a 30% infinite allele model (IAM). Since twelve polymorphic loci were used in the study, the Wilcoxon signed-rank test was suited for data analysis [42].

#### 2.2.3. ABC Analyses Using DIYABC

The demographic history of *O. hainanensis* was determined by approximate Bayesian computation (ABC) methods using DIYABC v.2.0.4 [43]. ABC analysis performed the computations by combining both microsatellite DNA and mtDNA data. The possible demographic history of *O. hainanensis* was tested in two steps (Figure 2). First, three potential scenarios were performed to test the changes in population sizes of *O. hainanensis*, assuming different ancestral (Ne) and current (Na) effective population sizes. These were: (1) a stable size population (Ne = Na), (2) a population expansion (Ne < Na), and (3) a population decline (Ne > Na). Second, based on the most appropriate scenario in the first step, the population size change always reflects the complex history, and the populations might have suffered a bottleneck event followed by a recent population expansion. Therefore, two competing demographic scenarios in ABC 2 analyses were constructed. Scenario A is a stable size population, Ne (long-term historical Ne), and scenario B (bottleneck and expansion model), the populations suffered a bottleneck event followed by a recent population expansion and the effective population size changed during t1 and t2 time. Following the preliminary analysis, three million datasets were simulated to generate the reference table, and all the summary statistics included in the software DIYABC were used [43]. The posterior probability of each model based on 1% of the simulated datasets for each scenario was assessed using both direct and logistic approaches [43].

## 3. Results

### 3.1. Mitochondrial DNA and Nuclear DNA Analysis

A total of 117 haplotypes were identified by sequencing 2066 bp of the complete mtDNA cyt *b* gene (1140 bp) and D-loop (926 bp) from 305 *O. hainanensis* individuals, with nine shared haplotypes in the total population (Table 1). The most common haplotypes were shared by three populations, in which populations BM, CJ, and ND shared H07 and populations LG, ND, and WQ shared H64. Six shared haplotypes are indicated in two populations, in which populations BM and ND share H06, H08, and H10, populations LOS and LW share H66 and H68, populations CJ and ND share H38, and populations LS and TQ share H71. The average haplotype diversity was high (0.984), ranging from 0.000 (TR) to 0.978 (CJ), the nucleotide diversity (θπ) within *O. hainanensis* was low (0.0107), ranging from 0.0000 (TR) to 0.0107 (LG), and the nucleotide diversity (θω) was low (0.0157), ranging from 0.0000 (TR) to 0.0075 (CH) (Table 1). For the nuclear RAG1 gene (1506 bp), 64 haplotypes were obtained in the 305 samples, in which the derived number of private haplotypes for each population varied between 0 (WC) and 7 (ND). Overall, the average haplotype diversity (h) was 0.897, ranging from 0.095 (HH) to 1.000 (TQ). The nucleotide diversity (θπ) was 0.0019, ranging from 0.0000 (HH) to 0.0027 (NL), and the nucleotide diversity (θω) was 0.0107, ranging from 0.0000 (HH) to 0.0039 (NL) (Table 1).

The pairwise *F*_ST_ values ranged from −0.042 (between LOS and LW) to 0.995 (between TR and LW), with a mean value of 0.682 in mtDNA. The pairwise *F*_ST_ values between the populations were significantly different in all pairwise comparisons, except LOS and LW and LQ and WQ (Appendix A). Moreover, the pairwise *F*_ST_ values between populations on Hainan Island and mainland China were relatively large and significant (*F*_ST_ > 0.25, very great differentiation; [44]). For the nuclear RAG1 gene, the pairwise *F*_ST_ values ranged from −0.113 (between ZB and JR) to 0.959 (between HH and LW), with a mean value of 0.293. We used analysis of variance (AMOVA) to test the probable factors shaping genetic structure according to geographical barriers. For the mitochondrial DNA, the AMOVA results indicated that most of the genetic variation was among populations within groups, i.e., two groups (Scenario Ⅰ, 68.09%), three groups (Scenario Ⅱ, 54.43%), and four groups (Scenario Ⅲ, 63.59%) (Table 2). When the populations were divided into two groups (Scenario Ⅰ), three groups (Scenario Ⅱ), and four groups (Scenario Ⅲ), 1.24%, 17.14%, and 0.35% of the total variation was found among groups, respectively (Table 2). For the nuclear RAG1 gene, the AMOVA results indicated that most of the genetic variation was within populations, i.e., two groups (Scenario Ⅰ, 69.58%), three groups (Scenario Ⅱ, 69.56%), and four groups (Scenario Ⅲ, 74.25%) (Table 2). When the populations were divided into two groups (Scenario Ⅰ), three groups (Scenario Ⅱ), and four groups (Scenario Ⅲ), −0.11%, 3.46%, and 0.93% of the total variation was found among groups, respectively (Table 2).

For mitochondrial DNA, the topological relationships from the phylogenetic analysis based on mtDNA genes support the formation of five major lineages according to the distribution pattern from different populations (Figure 3). Lineage A1 was distributed in eight populations from western Hainan Island and three populations around the Gulf of Tonkin in mainland China. Lineage A2 was distributed in six populations from eastern Hainan Island and four populations around the South China Sea in mainland China. Lineage B was composed of the individuals from CH on Hainan Island and HH in the Red River, and lineage C was composed of only one population (CH). Lineages D and E were composed of the populations LW, LOS, and LG, WQ on southeastern Hainan Island, respectively (Figure 3). The results of the minimum spanning network were congruent with the phylogenetic reconstruction and showed five main clades, with clades B and E being located in the interior and the others being located at the tip (Figure 4). Clade A and Clade C are separated from Clade B by 20 and 30 steps, respectively, and Clade D is separated from Clade E by 19 steps. According to the network, the CH and HH populations were the ancestral populations. For the nuclear RAG1 gene, the phylogenetic relationship and networks showed no evidence of significant geographical structure corresponding to the sampling populations (Appendix A).

### 3.2. Historical Population Demography

The neutrality test calculated by Tajima’s D and Fu’s *F*s tests revealed negative but nonsignificant Tajima’s D values and significant negative Fu’s *F*s values for the total population in *O. hainanensis* (Tajima’s D, −0.973, *p* > 0.10; Fu’s *F*s, −37.383, *p* < 0.01). Fu’s *F*s is more sensitive than Tajima’s D in the detection of population growth [45]. The Bayesian skyline plot showed that *O. hainanensis* populations appeared to remain stable over a long period and experienced continuous population growth beginning approximately 80,000 years ago and later expanding in a considerably fast manner in 50,000 years ago (Figure 5). A higher θπ than θω usually indicates population growth; otherwise, it reveals population decline (if θπ < θω). However, comparing current and historical genetic diversity (θω (0.015) > θπ (0.010)) indicated that the population of *O. hainanensis* showed a pattern of decline [21]. The dating analyses revealed that the four major lineages diverged at approximately 1.409 Ma (95% CI = 1.128–1.705 Ma). The times for lineages A + B + C and A + B were 1.301 Mya (95% CI = 0.983–1.618 Mya) and 1.018 Mya (95% CI = 0.760–1.306 Mya), respectively. Our analysis of ancestral area reconstruction (BBM) showed that the common ancestor of *O. hainanensis* was inferred to be distributed on southwestern Hainan Island (Changhua River). With regard to lineage A, almost all specimens might have derived from CH, for node 159 with 38.35% marginal probability. The HH population likely came from CH, for node 170 with 80.63% marginal probability. Node 192 with 49.28% marginal probability of WQ indicated that the LG and LOW populations might originate from the WQ populations, which themselves might have previously come from CH (Figure 6).

### 3.3. Microsatellite DNA

Micro-Checker [35] indicated that there was no indication of scoring error due to stuttering or allelic dropout and null alleles. A total of 247 alleles were detected for the 12 polymorphic markers in 264 individuals. The number of alleles (N_A_), allelic richness (A_R_), observed heterozygosity (H_O_), expected heterozygosity (H_E_), and inbreeding coefficient (*F*_IS_) per population are given in Appendix A. The average number of alleles for each population ranged from 3.33 (FC) to 10.66 (HH) (average = 6.81). The average number of alleles (6.81) per population ranged from 3.33 (FC) to 10.66 (HH). The mean allelic richness (5.19) per population ranged from 3.02 (FC) to 6.81 (CH). The mean expected heterozygosity (0.68) ranged from 0.51 (FC) to 0.81 (CH), and the observed heterozygosity (0.52) ranged from 0.40 (FC) to 0.66 (TR) per population. The positive *F*_IS_ indicated that the heterozygote deficiencies of all populations ranged from 0.135 (NL) to 0.354 (LS) (Appendix A).

Characteristics of 12 microsatellite loci per locus in *O. hainanensis* are given in Appendix A. The number of alleles ranged from 12 (Loci-7) to 30 (Loci-12) (average = 20.58). The average allelic richness per locus was 8.33 and ranged from 5.381 (Loci-7) to 10.711 (Loci-12). The observed heterozygosity (H_O_) was 0.526 and ranged from 0.309 (Loci-1) to 0.687 (Loci-11), and the expected heterozygosity (H_E_) was 0.686 and ranged from 0.499 (Loci-1) to 0.782 (Loci-12). The mean value of *F*_IS_ was 0.241, and all studied microsatellite DNA markers had a positive *F*_IS_ value, indicating a deficiency of heterozygotes compared to that predicted by the HWE.

From the microsatellite DNA data, the pairwise *F*_ST_ values ranged from 0.007 (between LW and LOS) to 0.435 (between LW and FC), with a mean value of 0.218, and the *R*_ST_ values ranged from 0.045 (between LG and WQ) to 0.694 (between LW and JR), with a mean value of 0.374 (Appendix A). The pairwise *F*_ST_ values between the HH and other populations were relatively large in all pairwise comparisons (Appendix A). The AMOVA results indicated that most of the genetic variation was within the population, i.e., two groups (Scenario Ⅰ, 80.03%), three groups (Scenario Ⅱ, 79.89%), and four groups (Scenario Ⅲ, 80.65%) (Table 2). When the populations were divided into two groups (Scenario Ⅰ), three groups (Scenario Ⅱ), and four groups (Scenario Ⅲ), only 1.72%, 2.52%, and 0.15% of the total variation was found among the groups, respectively (Table 2).

Microsatellite DNA data were analyzed by using the model-based clustering algorithm implemented in the software STRUCTURE and assigning individuals to populations, identifying admixture proportions at the individual level. The most likely number of populations represented by the K-value needed to explain the observed genotypes. Our results from population structure analysis suggested that all fourteen populations were divided into two main genetic clusters consisting of populations in Hainan Island and mainland China (K = 2, Ln P(K) = −14498.125; Stdev Ln P(K) = 43.512; Delta K = 9.093) (Figure 7). The first two principal coordinates in principal component analysis (PCA) were performed to verify the relationship using the genetic distances among populations. The variances of the first and second principal components were 69.48% and 15.80%, respectively. The first two components explained 85.28% of the total variation in PCA analysis and indicated that the populations of *O. hainanensis* could be divided into two groups, which populations in mainland China (except NL and TR) belonged to one group, and the remaining populations belonged to the other group (Appendix A). Phylogenetic tree construction following unrooted NJ clustering algorithm using the microsatellite DNA markers showed three groups, including the Red River (HH), Hainan Island, and mainland China groups (Appendix A). The normal ‘L’ shaped distribution of mode-shift test in all fourteen populations suggested there was a stable population (Appendix A). No significant heterozygosity excess was observed by the Wilcoxon test under both TPM and SMM, indicating that genetic bottlenecks were not detected in *O. hainanensis* due to mutation-drift equilibrium. Therefore, the demographic history of *O. hainanensis* became even more complex, and *O. hainanensis* populations had not experienced a recent genetic bottleneck.

### 3.4. Approximate Bayesian Computation

We performed ABC analyses to determine the possible demographic history of *O. hainanensis*. In the ABC1 analyses, the “stable scenario” was highly favored (posterior probability = 0.9888 [0.9716, 1.0000] over the “expansion scenario” scenario (posterior probability = 0.0110 [0.0000, 0.0281]) and the “bottleneck scenario” scenario (posterior probability = 0.0002 [0.0000, 0.0005]). The DIY-ABC results shown that the effective population size of *O. hainanensis* was constant from the past to the present. At the same time, however, such conflicts between the DIY-ABC result and the results from other methods. Therefore, we make use of the competing demographic scenarios in the ABC 2 analyses. Scenario B (bottleneck and expansion model) had the highest posterior probability, which had very close to the maximum possible value of 1.0 (0.9784 [0.8539, 1.0000]). Our results suggested that the populations of *O. hainanensis* suffered a bottleneck event followed by a recent population expansion.

## 4. Discussion

### 4.1. Genetic Diversity

Maintaining levels of genetic diversity is the product of the long-term survival and fitness of species or populations, as it is important to provide the ability to adapt to environmental change. Haplotype and nucleotide diversity are important indicators of genetic variation based on mtDNA, and the result of *O. hainanensis* is consistent with previous studies in Hainan Island and mainland China, with overall high haplotype diversity (0.984) but low nucleotide diversity (0.010) (e.g., *S. argentatus*, [2]; *C. sinensis*, [4]; *Onychostoma lepturum*, [12] and *Tanichthys albonubes*, [47]). However, the nucleotide diversity was similar to the nucleotide diversity of some species distributed on Hainan Island (e.g., *A. normalis*, [9]; *G. orientalis*, [8]; *T. albonubes*, [47]) but much lower than that of other species (e.g., *O. lepturum*, [12]) (Table 1). In the nuclear RAG1 gene, the values of the average haplotype diversity (h) were similar and nucleotide diversity (θπ) was relatively lower than the values of cyprinid species in mainland China (e.g., *Abbottina rivularis* [48]; *T. albonubes* [47]). In microsatellite DNA, the values of the average number of alleles per population and genetic diversity were relatively lower than the values of related species on Hainan Island (e.g., *S. argentatus*, [1]; *Osteochilus salsburyi*, [49]; *G. orientalis*, [8]). However, the number of alleles of *O. hainanensis* was relatively lower than the average value in freshwater fishes (9.1 (±6.1) alleles) by meta-analysis of microsatellite polymorphisms [50]. The presence of significant heterozygote deficiencies in all populations could result from the following: inbreeding, nonrandom sampling (sampling bias), population subdivision (Walhund effect), or genetic drift [51]. In addition, the populations of *O. hainanensis* living in the middle and lower reaches have a relatively lower genetic diversity because of overexploitation, water pollution, flow modification, and habitat degradation [14]. The genetic diversity level of *O. hainanensis* on Hainan Island was higher than the genetic diversity level in mainland China and the Red River based on mitochondrial and microsatellite markers. In general, populations on the mainland possess higher genetic diversity than those on the islands because the effective population size and genetic diversity are generally considered to be positively correlated (e.g., *C. sinensis*, [4]; *R. ocellatus*, [6]). According to previous studies, *O. hainanensis* is distributed only in the southern region of the Yunkai-Shiwan Mountains region, suggesting a peripheral population on mainland China [5,10]. Our results show that the area of the drainage basin was higher with higher genetic diversity (e.g., Ne, h). Concerning relatively high levels of genetic diversity in *O. hainanensis* populations, some relatively larger drainages on Hainan Island reflective of a large effective population size might play an important role in harboring more genetic variants and accumulating more genetic diversity.

### 4.2. Population Structure

Our combined population genetic analyses of the mtDNA, nuDNA, and microsatellite datasets clearly show that genetic differentiation was high among the collection locations of *O. hainanensis*. The mean *F*_ST_ values among sampling sites based on mitochondrial, nuclear, and microsatellite data were 0.682, 0.293, and 0.218, respectively. These results suggest that restricted gene flow is occurring and that this lack of gene flow most likely results from physical isolation between the populations, reflecting the dendritic nature of river systems and their unidirectional flow [52]. MtDNA differentiation (*F*_ST_ value) was generally higher than the *F*_ST_ value estimated with nuclear DNA and microsatellite data. Large pairwise genetic differences were found among the LW and LOS populations and all the other populations. Populations LW and LOS are located in the southeastern part of the island and flow eastward to the South China Sea. According to the network and BBM analyses, the populations LOS and LW may have been colonized by founders from population WQ. We suggested that the effects of isolation and small population size were important factors for genetic differentiation in the populations (LW and LOS) [10]. Additionally, populations on Hainan Island generally exhibit lower genetic differentiation than adjacent mainland China, presumably due to the initial loss of diversity upon foundation and a more constrained population size following foundation [53]. Landscape features are known as the other factor influencing population genetic structure in *O. hainanensis*, such that regional populations in mainland China were constricted by geographic barriers and have thus undergone long independent evolutionary histories during subsequent climatic oscillations. Factors intrinsic to species can result in specific distribution patterns and genetic differentiation. In the present study, the genetic differentiation of *O. hainanensis* populations was lower than previous studies for other freshwater species found almost exclusively in fast-flowing waters towards headwaters of river drainages on Hainan Island (e.g., *O. lepturum*, [12]; *Micronoemacheilus pulcher*, [54]; *G. orientalis*, [8]) but similar to *A. normalis* [9]. Several previous studies estimated that the Wuzhishan and Yinggeling Mountain Ranges, Gulf of Tonkin, and Qiongzhou Strait were important barriers limiting gene exchange between populations on both sides. (e.g., *C. gachua*, [1]; *G. orientalis*, [8]; *O. lepturum*, [12]; *A. normalis*, [9]). However, based on our geographical groups (Scenarios Ⅰ, Ⅱ, and Ⅲ), AMOVA identified the lowest variations among the groups using mtDNA, nuDNA, and microsatellite data (Table 2). Our previous results revealed that the Wuzhishan and Yinggeling Mountain Ranges, Gulf of Tonkin, and Qiongzhou Strait were not vicariant barriers for *O. hainanensis* populations. In terms of habitat and specificity, *O. hainanensis* populations are found mostly in the lowlands, an area that has higher connectivity during lower sea levels or flooding [14,55]. However, AMOVA showed 17.14% genetic differences among groups in Scenario Ⅱ based on mtDNA, indicating that the Gulf of Tonkin is not a major geographic barrier among populations on Hainan Island and around the Gulf of Tonkin in mainland China. These results, similar to previous studies in *C. gachua*, support the occurrence of enabling mixing of populations across this ocean barrier (Gulf of Tonkin) [1].

### 4.3. Phylogeography of O. hainanensis

During Pleistocene glacial cycles, a period with sea-level fluctuations and land bridges as possible dispersal corridors exerted a more profound influence on the patterns of modern freshwater fish distribution between the island and mainland. Due to volcanism and sinking land at approximately 2–2.5 mya, Hainan Island was first isolated from mainland China by the current Gulf of Tonkin and the Qiongzhou Strait (e.g., [56,57]). A recent phylogeographic study on a widely distributed freshwater fishes found high genetic differentiation among Hainan Island and mainland China as a result of the Gulf of Tonkin and Qiongzhou Strait acting as barriers to gene flow (e.g., *O. bidens* [58]; *Opsariichthys* [5]; *G. orientalis*, [8]). However, in the present study, the phylogenetic relationship showed that lineage A1 was distributed around the Gulf of Tonkin, including FC, BL, and NL populations in mainland China and eight populations in western Hainan Island. Lineage A2 was found around the South China Sea, including MY, JR, and TR populations in mainland China and seven populations in eastern Hainan Island. These results, combined insights from past geological changes, could provide the basis for dispersal paths in *O. hainanensis*. During the ice age, the exposure of land due to the decrease in sea level reconnected river systems for the dispersal of freshwater species, including the Gulf of Tonkin, the Qiongzhou Strait, and Hainan Island, which became part of the coastal plains of the Asian continent. According to the Pleistocene connections of paleo-drainages because marine regressions have explained spatial patterns of genetic structure of freshwater fishes, western Hainan Island was expected to be closely related to the Gulf of Tonkin, and eastern Hainan Island would be closely related to the South China Sea. Similar scenarios have been proposed as likely explanations for the geographic patterns of other freshwater fishes, such as *M. pulcher* [54] and *Liniparhomaloptera disparis* [59]. According to the phylogenetic tree, the Red River (population HH) in mainland China and the Changhua River (population CH) on Hainan Island formed a monophyletic group (lineage B) and were located in the basal position of lineage A. Lineage C was only found in the Red River and located in the Tip in the network. In our previous study, we suggested that rising sea levels would flow into the Gulf of Tonkin from the south direction during glacial retraction (e.g., [1]). The Red River and Changhua River, belonging to the old Red River drainage system, located in the southern part of the Gulf of Tonkin, were the first populations to be geographically isolated after the retreat of glaciation. Many phylogeographical studies of freshwater fishes revealed that the Changhua River and Red River also found a close genetic relationship of populations between Hainan Island and mainland China [1,10]. Interestingly, Lineages D and E were distributed in the Longshou River (LOS), Longwei River (LW), Longgun River (LG), and Wanquan River (WQ) in the southeastern part of Hainan Island. During the most extreme sea-level retreat in the Pleistocene, the architecture of paleodrainages was composed of small and steep coastal drainages that flowed directly to the South China Sea on southeastern Hainan Island [13]. Our previous study showed the effects of isolation and small population size, suggesting that populations LOS, LW, and LG may have been colonized by founders from population WQ [10]. The formation of two clusters (K = 2) in the STRUCTURE analysis reinforces geographic isolation by the sea and would have generated the genetic structure of *O. hainanensis*, which reflects weak gene flows between mainland China and Hainan Island. The results of the BBM analysis indicated that possible ancestral populations of *O. hainanensis* were distributed along the Changhua River on Hainan Island. Two dispersal events occurred: the population in the Changhua River migrated to the Red River (Lineage B), and the populations in the South Hainan region moved northwards to the North Hainan region, although populations in Northwest Hainan Island dispersed to the populations around the Gulf of Tonkin (Lineage A1) and populations in Northeast Hainan Island dispersed to the populations in mainland China (Lineage A2) (Figure 8).

### 4.4. Demographic History of O. hainanensis

High haplotype diversity, low nucleotide diversity values, and star-like networks were observed that indicate *O. hainanensis* following a recent population demographic expansion and population range expansion [60]. In addition, Tajima’s D test and Fu’s *F*s analysis were used for neutral evolution, and Bayesian skyline plot analyses were increasingly used to reconstruct the historical demographic expansions of *O. hainanensis*. In this study, the significantly negative values of Fu’s *F*s tests and the nonsignificantly negative values of Tajima’s D test were investigated. Fu’s *F*s test is much more sensitive in detecting population growth than Tajima’s D test [28]. Bayesian skyline plot analyses of all populations indicated that *O. hainanensis* might follow the post-LGM (Last Glacial Maximum) expansion pattern, whose effective population size rapidly increased approximately 15 Kya. After the Last Glacial Maximum, we speculated that the warm climate was suitable for *O. hainanensis* survival and provided more suitable habitat for supporting the occurrence of a recent population expansion in mainland China. Population expansion that occurred in mainland China has been reported in freshwater fishes such as *Sinibrama macrops* [61], the sharpbelly *Hemiculter leucisculus* [62], and the rosy bitterling *R. ocellatus* [6]. A higher θω than θπ for *O. hainanensis* indicated population decline based on mitochondrial data. These results indicated that *O. hainanensis* experienced a complex demographic history, and we focused on the demographic changes in recent time frames in the ABC scenarios. However, the results of our ABC 1 analyses indicate that the population size was constant and conflict with the results from other methods. Therefore, we defined two biologically meaningful scenarios of the competing demographic history in ABC 2 analyses. Our results revealed that the populations of *O. hainanensis* suffered a bottleneck event followed by a recent population expansion. We suggest that *O. hainanensis* populations were found mostly in the lowlands and a lack of suitable freshwater habitat in southern mainland China and Hainan during the Last Interglacial, with the greatest range expansion during the Last Glacial Maximum (Figure 8).

## Figures and Tables

**Figure 1 genes-13-01908-f001:**
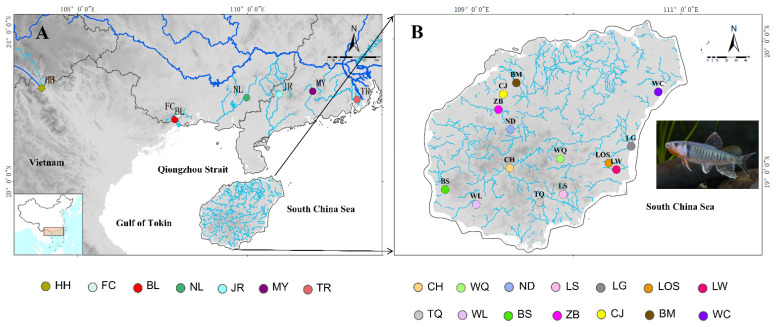
Maps of the study region in mainland China (**A**) and Hainan Island (**B**) indicate sites where 21 sampling localities of *O. hainanensis* were collected (circles). Sampling locations for the geotechnical sites were presented in the text and Table 1.

**Figure 2 genes-13-01908-f002:**
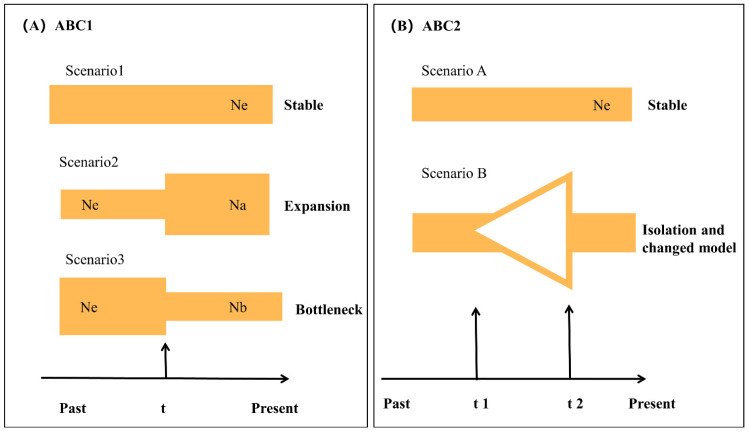
Schematic representation of five demographic scenarios for *O. hainanensis* tested by approximate Bayesian computation (ABC). Time and effective population size are not to scale. (**A**) Graphical representation of the three scenarios in ABC 1 analysis; (**B**) Graphical representation of the two scenarios in ABC 2 analysis.

**Figure 3 genes-13-01908-f003:**
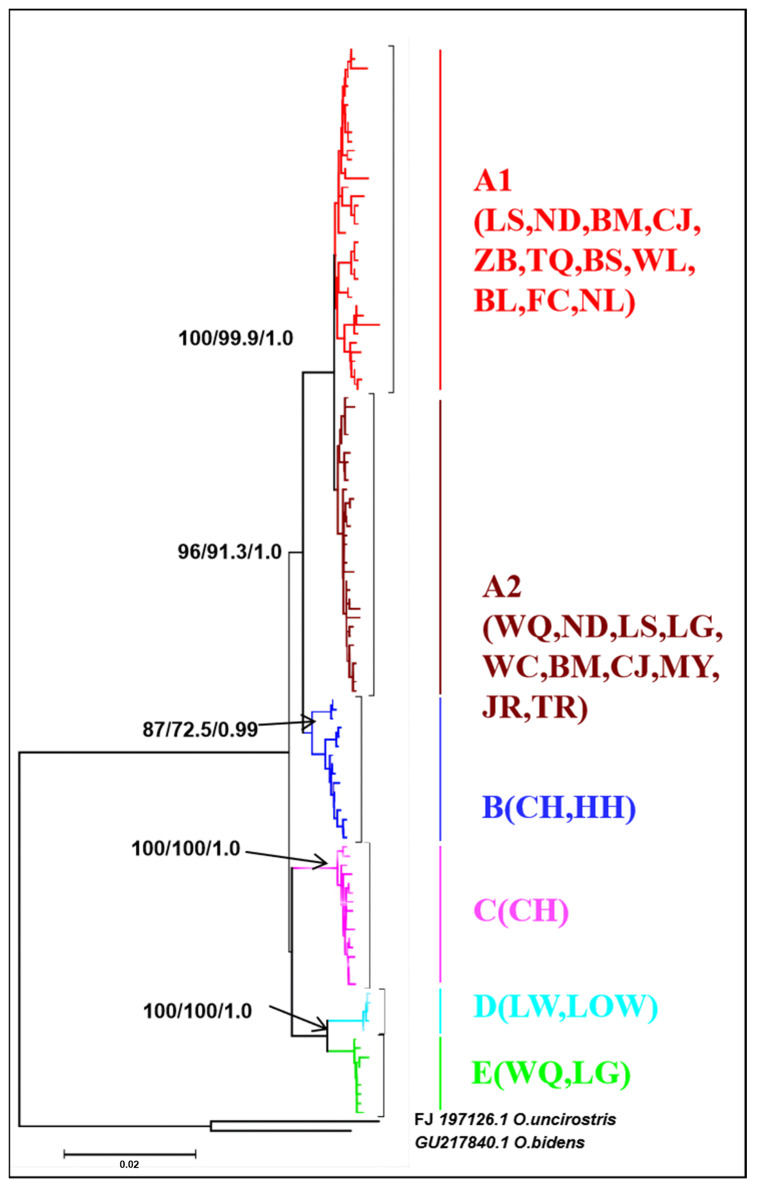
NJ tree of genetic relationships based on mitochondrial cyt *b* + D-loop gene among 21 populations in *O. hainanensis* using 117 haplotypes. The values above the branches are the posterior probabilities for bootstrap values for the NJ, ML, and Bayesian analyses. *O. bidens* and *O. uncirostris* are two species in the genus *Opsariichthys* used as an outgroup.

**Figure 4 genes-13-01908-f004:**
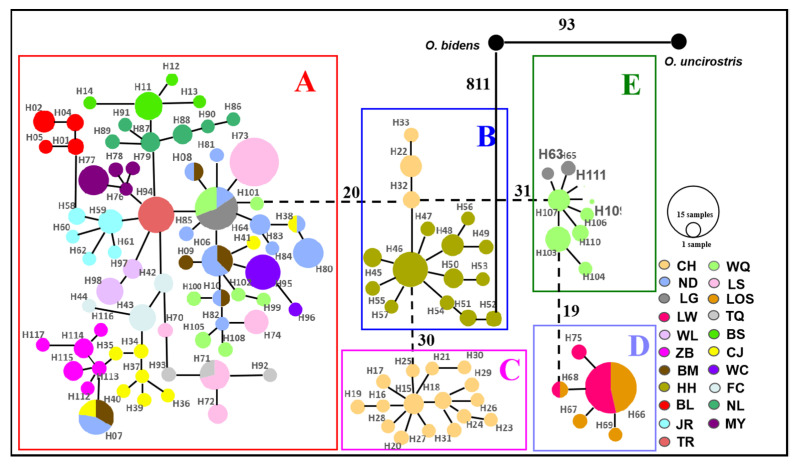
Minimum spanning network (MSN) of 117 haplotypes for *O. hainanensis* based on mitochondrial cyt *b* + D-loop genes identified in 305 individuals from 21 populations. Different colors indicate different sampling localities (see Figure 1). Circles represent haplotypes, and the proportion size is proportional to the number of individuals represented. Single lines directly connecting haplotypes indicate separation by one mutation step. The number of vertical bars on the connecting line is an increasing number of mutation steps. Clades A–E is shown on the map with corresponding colors for each lineage. *O. bidens* and *O. uncirostris* are two species in the genus *Opsariichthys* used as an outgroup.

**Figure 5 genes-13-01908-f005:**
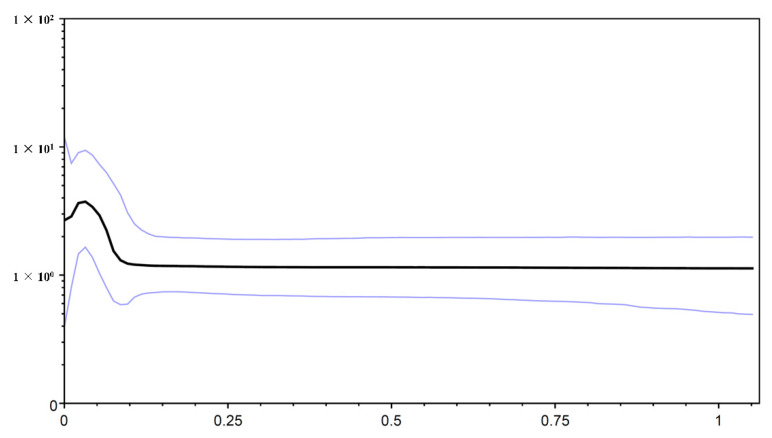
Bayesian skyline plots for the cyt *b* + D-loop of *O. hainanensis* from mainland China and Hainan Island. The *x*-axis gives units in millions of years before the present, and the *y*-axis is effective population size (Nes) and on a log scale. The solid line indicates the median estimate, whereas the thinner lines indicate the 95% credibility intervals.

**Figure 6 genes-13-01908-f006:**
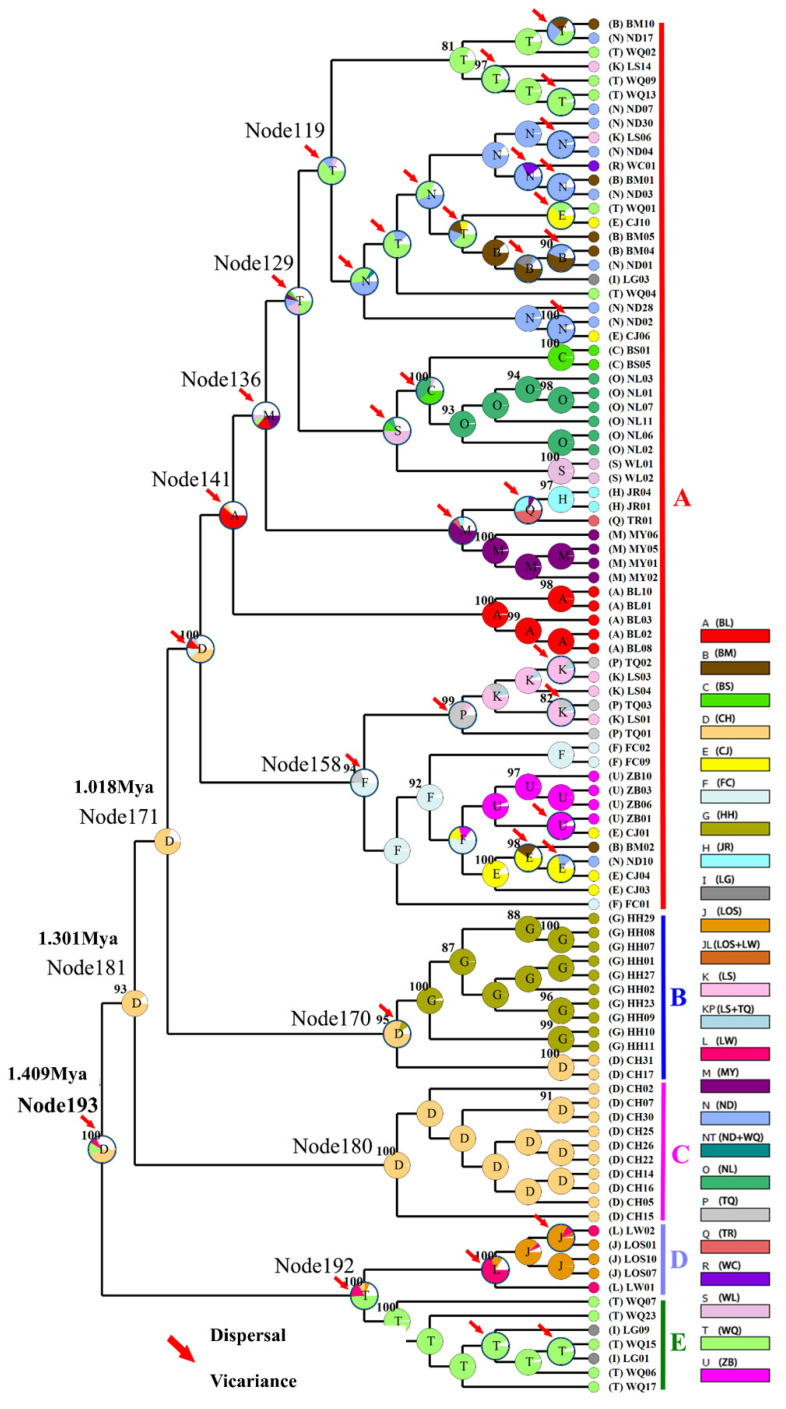
Ancestral states reconstructed by Bayesian binary MCMC (BBM) and plotted on the condensed tree calculated by RASP. The BBM ancestral area reconstructions with the highest likelihood are shown as pie charts for each *O. hainanensis* clade. The color key for ancestral reconstruction at nodes of interest obtained from BBM analysis is provided in the figure. The numbers in the circles represent the node number. Pie charts depict ancestral area reconstruction probability, with the colors of pie slices defined in the legend. A–E represent the different lineages.

**Figure 7 genes-13-01908-f007:**
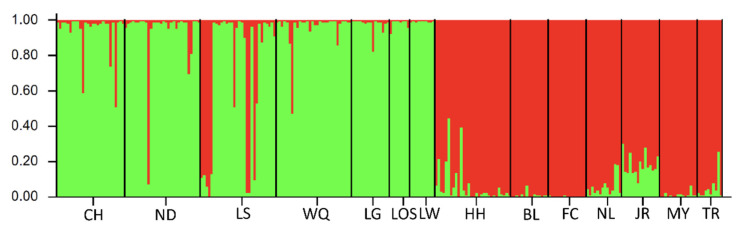
Results of STRUCTURE analysis for 264 *O. hainanensis* individuals based on microsatellite data. Estimated population structure for K = 2 using the method by Evano [46]. Each individual is represented by a vertical-colored bar, and the separation of the column into two colors represents the probability of membership in the relevant cluster. The *x*-axis represents individual samples and *y*-axis represents the proportion of belonging to a certain cluster. Black solid vertical lines separate each population whose names were indicated (code name given in Table 1).

**Figure 8 genes-13-01908-f008:**
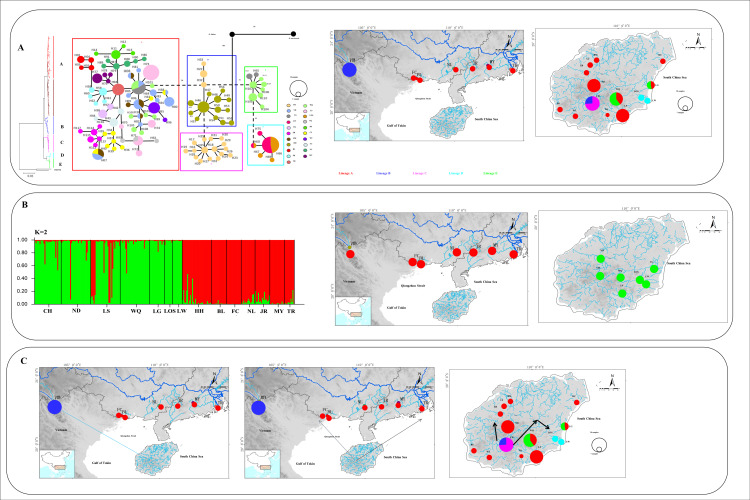
(**A**) Phylogenetic tree, the haplotype network, and the distribution of the major mtDNA lineages. (**B**) STRUCTURE based on microsatellite data and the distribution of groups (K = 2) revealed in the STRUCTURE. (**C**) The colonization history of *O. hainanensis*. Arrows represent migration routes. Lineages A–E is shown in the map with corresponding colors for each lineage.

**Table 1 genes-13-01908-t001:** Sampling localities, abbreviations, and genetic diversity indexes for *O. hainanensis* based on mitochondrial and nuclear data. N, sample size; Nh, haplotype numbers; *h*, haplotype diversity; π, θ, nucleotide diversity. Values in bold indicated geographic regions.

Gene	Mitochondrial DNA (cyt *b* Gene and d-Loop Region)					Nuclear DNA (RAG1)				
Locations (Abbreviation)	Sample Size (N)	Haplotype Numbers (N*h*)	Haplotype Diversity (*h*)	Nucleotide Diversity (π)	Nucleotide Diversity (θ)	Sample Size (N)	Haplotype Numbers (N*h*)	Haplotype Diversity (*h*)	Nucleotide Diversity (π)	Nucleotide Diversity (θ)
**Hainan Island**	**215**	**80**	**0.975**	**1.148**	**1.311**	**183**	**45**	**0.908**	**0.199**	**0.850**
Changhua River (CH)	27	19	0.960	0.742	0.728	24	6	0.725	0.250	0.445
Nandu River (ND)	30	12	0.897	0.204	0.244	20	13	0.947	0.185	0.281
Wanqan River (WQ)	30	14	0.924	1.062	0.635	24	11	0.902	0.157	0.249
Lingshui River (LS)	30	5	0.703	0.311	0.220	20	10	0.905	0.151	0.206
Longgun River (LG)	15	3	0.600	1.071	0.640	15	9	0.876	0.166	0.204
Longshou River (LOS)	10	4	0.533	0.029	0.051	9	5	0.722	0.125	0.171
Longwei River (LW)	10	3	0.378	0.019	0.034	10	2	0.200	0.013	0.023
Tengqiao River (TQ)	4	3	0.833	0.169	0.185	4	4	1.000	0.243	0.254
Wanglou River (WL)	9	2	0.500	0.024	0.018	8	5	0.893	0.175	0.154
Baisha River (BS)	10	4	0.533	0.106	0.188	10	3	0.378	0.027	0.047
Zhubi River (ZB)	10	6	0.867	0.110	0.137	10	3	0.378	0.134	0.164
Chunjiang River (CJ)	10	9	0.978	0.314	0.342	9	5	0.861	0.148	0.147
Beimen River (BM)	10	5	0.844	0.267	0.222	10	6	0.889	0.174	0.117
Wenchang River (WC)	10	2	0.200	0.029	0.051	10	2	0.467	0.031	0.023
**Mainland China**	**90**	**37**	**0.960**	**0.737**	**0.754**	**74**	**19**	**0.817**	**0.172**	**0.463**
Red River (HH)	30	13	0.885	0.184	0.244	21	2	0.095	0.006	0.018
Beilun River (BL)	10	5	0.822	0.090	0.086	10	5	0.822	0.099	0.094
Fangheng River (FC)	10	3	0.600	0.158	0.188	10	3	0.378	0.053	0.094
Nanliu River (NL)	10	6	0.867	0.125	0.154	10	7	0.911	0.274	0.399
Jian River (JR)	10	5	0.756	0.075	0.120	5	3	0.700	0.186	0.223
Moyang River (MY)	10	4	0.533	0.065	0.068	10	4	0.533	0.103	0.164
Tan River (TR)	10	1	0.000	0.000	0.000	8	4	0.821	0.164	0.128
**Total**	305	117	0.984	1.076	1.576	257	59	0.897	0.194	1.073

**Table 2 genes-13-01908-t002:** Analysis of molecular variance (AMOVA) for O. hainanensis based on mitochondrial, nuclear, and microsatellite data.

	Mitochondrial cyt *b* + D-Loop Gene			nuDNA RAG1 Genes			Microsatellite DNA		
Scenario Ⅰ: two independent groups divided by the Qiongzhou Strait									
Among groups	1.24	*F*_CT_ = 0.012	0.292	−0.11	*F*_CT_ = −0.001	0.317	1.72	*F*_CT_ = 0.017	0.004
Among populations within groups	68.09	*F*_SC_ = 0.690	0.000	30.53	*F*_SC_ = 0.305	0.000	18.24	*F*_SC_ = 0.186	0.000
Within populations	30.66	*F*_ST_ = 0.693	0.000	69.58	*F*_ST_ = 0.304	0.000	80.03	*F*_ST_ = 0.200	0.000
Scenario Ⅱ: three independent groups divided by the Qiongzhou Strait and Beibu Gulf;									
Among groups	17.14	*F*_CT_ = 0.171	0.242	3.46	*F*_CT_ = 0.035	0.389	3.09	*F*_CT_ = 0.031	0.370
Among populations within groups	54.43	*F*_SC_ = 0.657	0.000	26.98	*F*_SC_ = 0.279	0.000	12.31	*F*_SC_ = 0.127	0.000
Within populations	28.43	*F*_ST_ = 0.716	0.000	69.56	*F*_ST_ = 0.304	0.000	84.6	*F*_ST_ =0.154	0.000
Scenario Ⅲ: four groups divided by the WY Range, the Qiongzhou Strait and Beibu Gulf.									
Among groups	0.35	*F*_CT_ = 0.004	0.317	0.93	*F*_CT_ = 0.009	0.266	−1.95	*F*_CT_ = −0.019	0.635
Among populations within groups	63.59	*F*_SC_ = 0.638	0.000	24.82	*F*_SC_ = 0.251	0.000	13.99	*F*_SC_ = 0.137	0.000
Within populations	36.06	*F*_ST_ = 0.639	0.000	74.25	*F*_ST_ =0.258	0.000	87.96	*F*_ST_ = 0.120	0.000
SAMOVA									
Among groups	63.12	*F*_CT_ = 0.631	0.000	31.2	*F*_CT_ = 0.312	0.000	14.93	*F*_CT_ = 0.149	0.000
Among populations within groups	7.59	*F*_SC_ = 0.206	0.000	11.45	*F*_SC_ = 0.166	0.000	10.04	*F*_SC_ = 0.118	0.000
Within populations	29.28	*F*_ST_ = 0.707	0.000	57.35	*F*_ST_ = 0.426	0.000	75.03	*F*_ST_ = 0.250	0.000

## Data Availability

The data that support the findings of this study are available in GenBank (https://www.ncbi.nlm.nih.gov/, accessed on 1 June 2022), reference number MN325030-MN325066.

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
