# Peer review of "Multilocus Phylogeography and Population Genetic Analyses of *Opsariichthys hainanensis* Reveal Pleistocene Isolation Followed by High Gene Flow around the Gulf of Tonkin"

_genes, 2022, doi:10.3390/genes13101908_

Round 1

Reviewer 1 Report

This report covers important new ground for resolving the complex historical phylogeography and population demographics unique to Hainan Island freshwater fishes in relation to mainland China and northern Vietnam.  As such it warrants publication in Genes as an important regional reference for sustained management and protection of evolutionary biodiversity in an area of progressive habitat loss and disturbance due to anthropogenic impacts.  

The data are extensive and sufficient and rigorously analyzed using a diversity of standard well established statistical workflows.  All methods are implemented in an appropriate and authoritative manner.  

The manuscript would benefit by integrating the different categorical results across biomarkers into a summary Figure 8 that illustrates the main historical demographic scenario described near the end of the Discussion and proposed by the authors as reflected by the title of the study.  This should be mapped in historical detail onto the geographic region of Hainan vs. the continental mainland and not just be summarized as an abstract conceptual diagram as in Fig. 2.

The authors should speak more directly to the possible sources of error leading to inconsistent results and possible alternative explanations for interpreting the same original data analyzed under a Bayesian framework. In particular, conflicting results in the ABC 1 vs 2 analyses make it difficult to interpret how Fig 6 and Fig 7, for instance, inform the main synthesis of demographic results and conclusions of the research. This would not only strengthen the discussion but make the final visual summary in Figure 8 more accessible and perhaps more convincing to readers.

Minor comments:

"Fish" denotes multiple individuals of the same species. "Fishes" denotes multiple individuals across multiple species. The authors need to correct for consistent usage of fish vs fishes throughout the entire ms.  For instance, Abstract line 20 and Introduction line 38 "fishes" is correct but then is not correct in Intro lines 40, 43, 44...  Correct again on line 51 but incorrect on line 55, etc.

NJ tree figures in SuppMat without any bootstrap confidence intervals are essentially meaningless from a statistical standpoint.  

It would also be good to present somewhere (perhaps in SuppMat) a table of the exact values corresponding to the bubble diameters and branch distances plotted in the haplotype networks.

Author Response

Reviewer 1

Comments and Suggestions for Authors

 This report covers important new ground for resolving the complex historical phylogeography and population demographics unique to Hainan Island freshwater fishes in relation to mainland China and northern Vietnam.  As such it warrants publication in Genes as an important regional reference for sustained management and protection of evolutionary biodiversity in an area of progressive habitat loss and disturbance due to anthropogenic impacts.  

The data are extensive and sufficient and rigorously analyzed using a diversity of standard well established statistical workflows.  All methods are implemented in an appropriate and authoritative manner.  

The manuscript would benefit by integrating the different categorical results across biomarkers into a summary Figure 8 that illustrates the main historical demographic scenario described near the end of the Discussion and proposed by the authors as reflected by the title of the study.  This should be mapped in historical detail onto the geographic region of Hainan vs. the continental mainland and not just be summarized as an abstract conceptual diagram as in Fig. 2.

The authors should speak more directly to the possible sources of error leading to inconsistent results and possible alternative explanations for interpreting the same original data analyzed under a Bayesian framework. In particular, conflicting results in the ABC 1 vs 2 analyses make it difficult to interpret how Fig 6 and Fig 7, for instance, inform the main synthesis of demographic results and conclusions of the research. This would not only strengthen the discussion but make the final visual summary in Figure 8 more accessible and perhaps more convincing to readers.

→As requested, we added Figure 8 and revised it. “Figure 8. (A) Phylogenetic tree, the haplotype network, and the distribution of the major mtDNA lineages. (B) STRUCTURE based on microsatellite data and the distribution of groups (K=2) revealed in the STRUCTURE. (C) The colonization history of Opsariichthys hainanensis. Arrows represent migration routes.”

Minor comments:

"Fish" denotes multiple individuals of the same species. "Fishes" denotes multiple individuals across multiple species. The authors need to correct for consistent usage of fish vs fishes throughout the entire ms.  For instance, Abstract line 20 and Introduction line 38 "fishes" is correct but then is not correct in Intro lines 40, 43, 44...  Correct again on line 51 but incorrect on line 55, etc.

→ As requested, we revised it.

NJ tree figures in SuppMat without any bootstrap confidence intervals are essentially meaningless from a statistical standpoint.  

 →As requested, we added the bootstrap values and revised them.

It would also be good to present somewhere (perhaps in SuppMat) a table of the exact values corresponding to the bubble diameters and branch distances plotted in the haplotype networks.

→ As requested, we added the bubble diameters and revised them in the haplotype networks. 

Submission Date

31 August 2022

Date of this review

13 Sep 2022 00:56:21

Reviewer 2 Report

Major comments:

This paper reports genetic characteristics and phylogeography of Opsariichthys hainanensis in Hainan Island and mainland China by analyzing three kinds of DNA (cytb gene and D-loop in mtDNA, nuclear RAG1 gene and 12 microsatellites). Although this paper is well written, it wears many mistakes in data analysis and interpretation. So, they must be corrected, prior to acceptance. They are as follows.

1.    Molecular clock (line 181): How did the authors decide the molecular clock (i.e., mutation rate) of mtDNA in O. hainanensis? This is very important for BSP and the estimation of divergence time of clades. In my opinion, the clock rate (5% in D-loop and 1.05% in cyt b) is too high for cyprinid fishes. This greatly affects Fig. 5.

2.    Outgroup: Fig. 3 is lacking in outgroup. Therefore, the order of divergence is incorrect. This should be redrawn with proper outgroup species. The same also applied to Fig. 6 and Fig. S4. Fig. S4 should be redrawn as unrooted tree with bootstrap values.

3.    Neutrality test: In Fu’s neutrality test, the critical value of significance is 0.02, not 0.05. See the manual of Arlequin.

4.    Bottleneck test (Table S4): The mutation model of microsatellite is generally explained with both IMM and SMM (that is, TPM). In TPM, the weight of SMM is much larger than that of IMM. Therefore, the results of Wilcoxon’s test mean no sigh of bottleneck in all the populations. In addition, mode-shift indicator is very sensitive to bottlenecks in the past. A normal L-shaped allele frequency distribution strongly supports no bottleneck (L411-420). The populations used in this study should be considered not to have experienced bottlenecks in the past.

5.    Nuclear RAG1 gene: How do the authors explain genetic diversity of RAG1 gene in the phylogeography of O. hainanensis?

The authors conclude that populations of O. hainanensis in mainland China dispersed from Hainan Island and the populations suffered from bottlenecks in the past. However, in the analysis without outgroup, this dispersal hypothesis is not supported. On the contrary, the reverse process (from mainland China to Hainan Island) is possible. In addition, the results of bottleneck test means no bottleneck in the past.

Minor comments:

Text

Line 86: Citation no. 1 of Zhang et al. is correct?

Line 262: Haplotype ID of mtDNA should be changed in order to escape the confusion of that of RAG1 haplotype. The same ID is observable in Fig. 4 and Fig. S2.

Lines 289-297: Some data are different from those of Table 2. Check the values again.

Lines 316-317: In the analysis without outgroup, this is unbelievable.

Liens 341-343: The result of dating analysis should be shown in molecular tree. In addition, the methodology of this analysis is not described in M&M. This is important.

Line 408: In Fig. S3, populations in mainland China, except NL and TR, does not seem to be composed of one cluster.

Line 410: In my eye, three groups are not recognized.

Lines 430-435: The result of ABC analysis seems to well supports those of bottleneck test in lines 411-420.

Figure and Table

Figure 1: (A) and (B) should be added to figure.

Table 1: In column 1, Hainan island and mainland China is to be separated from their populations.

Figure S1: What do red triangles mean?

Figure S3: In this figure, percentage values of components are not described.

Figure S4: This figure should be redrawn as unrooted tree with bootstrap values.

Table S1: Is this table based on RYR3 gene, not RAG1 gene?

Table S2: Allelic richness is estimated, based on the smallest number of populations. In this table, this is on 8 individuals of population LOS. This number is too small to grasp at genetic diversity of populations correctly.

Author Response

Reviewer 2

Comments and Suggestions for Authors

Major comments:

This paper reports genetic characteristics and phylogeography of Opsariichthys hainanensis in Hainan Island and mainland China by analyzing three kinds of DNA (cytb gene and D-loop in mtDNA, nuclear RAG1 gene and 12 microsatellites). Although this paper is well written, it wears many mistakes in data analysis and interpretation. So, they must be corrected, prior to acceptance. They are as follows.

  1. Molecular clock (line 181): How did the authors decide the molecular clock (i.e., mutation rate) of mtDNA in O. hainanensis? This is very important for BSP and the estimation of divergence time of clades. In my opinion, the clock rate (5% in D-loop and 1.05% in cyt b) is too high for cyprinid fishes. This greatly affects Fig. 5.

→As requested, we revised it. “In this study, mutation rates for D-loop and cyt b were regarded as 3.6% and 0.76% per million years in cyprinid fishes for population expansion, respectively.”

D-loop 3.6%, Donaldson and Wilson, 1999

Donaldson, K. A., & Wilson Jr, R. R. (1999). Amphi-panamic geminates of snook (Percoidei: Centropomidae) provide a calibration of the divergence rate in the mitochondrial DNA control region of fishes. Molecular phylogenetics and evolution13(1), 208-213.

cyt b 0.76%, Zardoya and Doadrio 1999

Zardoya, R., & Doadrio, I. (1999). Molecular evidence on the evolutionary and biogeographical patterns of European cyprinids. Journal of molecular evolution, 49(2), 227-237.

  1. Outgroup: Fig. 3 is lacking in outgroup. Therefore, the order of divergence is incorrect. This should be redrawn with proper outgroup species. The same also applied to Fig. 6 and Fig. S4. Fig. S4 should be redrawn as unrooted tree with bootstrap values.

 →As requested, we revised it. Opsariichthys bidens and O. uncirostris are two species in the genus Opsariichthys used as an out-group in Fig. 3. Ancestral states reconstructed by Bayesian binary MCMC (BBM) and plotted on the condensed tree calculated by RASP in Fig. 6. In general, this analysis does not include outgroups. We added the bootstrap values in Fig. S4.

  1. Neutrality test: In Fu’s neutrality test, the critical value of significance is 0.02, not 0.05. See the manual of Arlequin.

→As requested, we revised it.

  1. Bottleneck test (Table S4): The mutation model of microsatellite is generally explained with both IMM and SMM (that is, TPM). In TPM, the weight of SMM is much larger than that of IMM. Therefore, the results of Wilcoxon’s test mean no sigh of bottleneck in all the populations. In addition, mode-shift indicator is very sensitive to bottlenecks in the past. A normal L-shaped allele frequency distribution strongly supports no bottleneck (L411-420). The populations used in this study should be considered not to have experienced bottlenecks in the past.

→As requested, we agreed that the results of bottleneck test means no bottleneck in the past and revised it. But a higher θω than θπ for O. hainanensis indicated population decline based on mitochondrial data. We suggested that O. hainanensis experienced a complex demographic history.

  1. Nuclear RAG1 gene: How do the authors explain genetic diversity of RAG1 gene in the phylogeography of O. hainanensis?

→As requested, we revised it. “In the nuclear RAG1 gene, the values of the average haplotype diversity (h) were similar and nucleotide diversity (θπ) was relatively lower than the values of cyprinid species in mainland China (e.g., Abbottina rivularis (Jang-Liaw et al., 2019); Tanichthys albonubes (Zhao et al., 2018))”  

Jang-Liaw, N. H., Tominaga, K., Zhang, C., Zhao, Y., Nakajima, J., Onikura, N., & Watanabe, K. (2019). Phylogeography of the Chinese false gudgeon, Abbottina rivularis, in East Asia, with special reference to the origin and artificial disturbance of Japanese populations. Ichthyological Research, 66(4), 460-478.

Zhao, J., Hsu, K. C., Luo, J. Z., Wang, C. H., Chan, B. P., Li, J., ... & Lin, H. D. (2018). Genetic diversity and population history of Tanichthys albonubes (Teleostei: Cyprinidae): Implications for conservation. Aquatic Conservation: Marine and Freshwater Ecosystems, 28(2), 422-434.

The authors conclude that populations of O. hainanensis in mainland China dispersed from Hainan Island and the populations suffered from bottlenecks in the past. However, in the analysis without outgroup, this dispersal hypothesis is not supported. On the contrary, the reverse process (from mainland China to Hainan Island) is possible. In addition, the results of bottleneck test mean no bottleneck in the past.

→As requested, we added the outgroup and revised it. The outgroups connected to the HH population in the network revealed that the CH and HH populations were the ancestral populations. According to the network, clades B (the CH and HH populations) was located in the interior and indicated this was the ancestral clade. Furthermore, the common ancestor of O. hainanensis was inferred to be distributed on southwestern Hainan Island (Changhua River) based on the ancestral area reconstruction (BBM) analysis. As requested, we agreed that the results of bottleneck test mean no bottleneck in the past and revised it. But a higher θω than θπ for O. hainanensis indicated population decline based on mitochondrial data. We suggested that O. hainanensis experienced a complex demographic history.

Minor comments:

Text

Line 86: Citation no. 1 of Zhang et al. is correct?

→As requested, we revised it. “Zhang et al. [10]”

Line 262: Haplotype ID of mtDNA should be changed in order to escape the confusion of that of RAG1 haplotype. The same ID is observable in Fig. 4 and Fig. S2.

→As requested, we revised it. For the nuclear RAG1 gene, we denote by HN in Fig. S2.

Lines 289-297: Some data are different from those of Table 2. Check the values again.

→As requested, we checked and revised it.

Lines 316-317: In the analysis without outgroup, this is unbelievable.

→As requested, we added the outgroup and revised it. The outgroups connected to the HH population in the network revealed that the CH and HH populations were the ancestral populations.

Liens 341-343: The result of dating analysis should be shown in molecular tree. In addition, the methodology of this analysis is not described in M&M. This is important.

→As requested, we added and revised it in M&M. “In addition, we estimate a time period for the most recent common ancestor (TMRCA) for each lineage as implemented in BEAST v1.8.2 software (Drummond et al., 2012). In this study, Divergence times were estimated under a strict molecular clock (uncorre-lated lognormal), and mutation rates for D-loop and cyt b were regarded as 3.6% and 0.76% per million years in cyprinid fishes, respectively.”

Line 408: In Fig. S3, populations in mainland China, except NL and TR, does not seem to be composed of one cluster.

→As requested, we added a circle to mark the same group in Fig. S3.

Line 410: In my eye, three groups are not recognized.

→As requested, we revised it. “The first two components explained 85.28% of the total variation in PCA analysis and indicated that the populations of O. hainanensis could be divided into two groups, which populations in mainland China (except NL and TR) belonged to one group, and the remaining populations belonged to the other group (Figure S3).”

Lines 430-435: The result of ABC analysis seems to well support those of bottleneck test in lines 411-420.

→As requested, we agreed that the results of bottleneck test means no bottleneck in the past and revised it. But, a higher θω than θπ for O. hainanensis indicated population decline based on mitochondrial data. We suggested that O. hainanensis experienced a complex demographic history.

Figure and Table

Figure 1: (A) and (B) should be added to figure.

→As requested, we added the marker and revised it.

Table 1: In column 1, Hainan island and mainland China is to be separated from their populations.

→As requested, we boldface to compartmentalize both.

Figure S1: What do red triangles mean?

→As requested, we added and revised it. “Red triangles mean shared haplotype.”

Figure S3: In this figure, percentage values of components are not described.

→As requested, we added the percentage values and revised them.

Figure S4: This figure should be redrawn as unrooted tree with bootstrap values.

→As requested, we added the bootstrap values and revised them.

Table S1: Is this table based on the RYR3 gene, not RAG1 gene?

 →As requested, we revised it.

Table S2: Allelic richness is estimated, based on the smallest number of populations. In this table, this is on 8 individuals of population LOS. This number is too small to grasp at the genetic diversity of populations correctly.

→Longshou River (LOS) is the smallest of the rivers we collect on Hainan Island. in Hainan Island, pollution might extremely threaten the survival of freshwater fishes. We're sorry that only eight individuals were collected from the Longshou River (LOS). But, the allelic richness of Longshou River (LOS) is not the smallest number of all populations. So, we don't think this result will affect the overall conclusion.

Submission Date

31 August 2022

Date of this review

26 Sep 2022 15:11:14
